# Unpacking the Lay Epidemiology of Cervical Cancer: A Focus Group Study on the Perceptions of Cervical Cancer and Its Prevention among Women Late for Screening in Norway

**DOI:** 10.3390/healthcare11101441

**Published:** 2023-05-15

**Authors:** Gunvor Aasbø, Bo T. Hansen, Jo Waller, Mari Nygård, Kari N. Solbrække

**Affiliations:** 1Department of Research, Cancer Registry of Norway, 0304 Oslo, Norway; 2Department of Interdisciplinary Health Science, Institute of Health and Society, University of Oslo, 0318 Oslo, Norway; 3Department of Infection Control and Vaccine, Norwegian Institute of Public Health, 0213 Oslo, Norway; 4School of Cancer and Pharmaceutical Sciences, King’s College London, London WC2R 2LS, UK

**Keywords:** cervical cancer screening, nonattendance, lay epidemiology, lay perspective, candidacy, aetiology, uncertainty, risks, Norway

## Abstract

Nonattendance for cervical cancer screening is often understood in terms of a lack of ‘appropriate’ or ‘correct’ knowledge about the risks and prevention of the disease. Few studies have explored how lay persons—the users themselves—interpret and contextualise scientific knowledge about cervical cancer. In this study, we address the following research question: How is the epidemiology of cervical cancer and its prevention discussed among women who are late for cervical cancer screening in Norway? We completed nine focus group interviews (FGIs) with 41 women who had postponed cervical screening. The analyses were both inductive and explorative, aiming to unpack the complexity of lay understandings of cervical cancer. Interactive associations expressed in the FGIs reflected multiple understandings of aetiology and risk factors, screening, and interpretations of responsibility for acquiring cervical cancer. The term ‘candidacy’ was employed to provide an enhanced understanding of lay reasoning about the explanations and predictions of cervical cancer, as reflected in the FGIs. Both interpretations of biomedical concepts and cultural values were used to negotiate acceptable and nuanced interpretations of candidacy for cervical cancer. Uncertainties about risk factors for acquiring cervical cancer was an important aspect of such negotiations. The study’s findings provide an in-depth understanding of the contexts in which screening may be rendered less relevant or significant for maintaining health. Lay epidemiology should not be considered inappropriate knowledge but rather as a productive component when understanding health behaviours, such as screening attendance.

## 1. Introduction

Cervical cancer is the fourth most common cancer among women worldwide [1]. Human papillomavirus (HPV) is sexually transmitted, and persistent infection with high-risk HPV is responsible for nearly all cervical cancer cases [2].

The incidence [3,4] and mortality [5] of cervical cancer may be substantially reduced by organised screening. Today, more than half of all invasive cervical cancers diagnosed in countries with organised screening programmes occur in underscreened or unscreened women [6]. Thus, screening nonattendance is associated with a relatively high risk of cervical cancer. Although the implementation of the HPV vaccination is likely to contribute tomore efficient prevention of cervical cancer in the future [7], screening will remain an important public health intervention for many years [8].

In Norway, all women aged 25–69 years are enrolled in the National Cervical Cancer Screening Programme (NCCSP). The implementation of HPV testing as primary screening has been ongoing for the last five years, and since the summer of 2022, it has been implemented nationwide and replaced cytology-based screening. Thus, the recommended screening interval has changed from every three years to every five years. The NCCSP uses a centralised invitation procedure to issue standard invitations to women who have not been registered for a screening test during the recommended interval, encouraging them to schedule a screening appointment with their general practitioner (GP) or gynaecologist. If a woman is still not registered for a screening test the following year, she receives a reminder letter. The NCCSP sends out letters once a month. The screening sample is collected through a gynaecological examination. Positive screening tests are triage-tested and do not require a new screening sample. A complex algorithm (Appendix A) determines the further follow-up care of women. The NCCSP aims for an 80% overall attendance rate, but it has stagnated at around 70% [9]. The participation rates dropped during COVID-19 but have now stabilised at the pre-COVID level [9]. A recent study showed that HPV self-sampling is likely to increase attendance for cervical screening in Norway [10], but HPV self-sampling is not yet offered by the NCCSP.

Previous studies on nonattendance in organised cervical cancer screening programmes in Norway and internationally point to practical and emotional barriers [11,12], interpretations of invitation strategies for cervical cancer screening [13], and questioning the relevance and value of screening [11]. Awareness of screening, knowledge of HPV, and related health beliefs, such as cancer fatalism and individual risk perceptions, have also been found to be associated with cervical screening attendance [14,15]. However, research on health behaviours and beliefs has not typically addressed how people contextualise and interpret epidemiology [16,17]. For instance, as a disease caused by a sexually transmitted virus, cervical cancer may be associated with a set of moral understandings. Hence, the boundaries between risk factors for cervical cancer and moralisation about behaviours associated with the disease may blur [16]. Lay interpretations of the epidemiology of cervical cancer may provide insights into how scientific knowledge is evaluated, given legitimacy, or (de)valued. The purpose of this study is to unpack lay interpretations of scientific knowledge to generate an enhanced understanding of the contexts in which the relevance of screening and decision-making with regard to screening attendance are played out.

### 1.1. Theoretical Approach

Within the field of medical sociology, lay beliefs about health and disease have been analysed and theorised over the last four decades by embracing variety, complexity, and multi-functionality [18,19]. This interest includes a strong focus on the relationship between the prevention of disease and its inevitability [20]. The notion of ‘lay epidemiology’ refers to knowledge and beliefs about health, the causation of disease, and health risks based on non-traditional sources of information [20,21,22], as well as selectively incorporated, adapted, and supplemented scientific medical understandings [23]. To unpack the lay epidemiology of cervical cancer, we specifically draw on the term ‘candidacy’, coined by Davison et al. [20], which refers to the idea of ‘the kind of person who gets heart trouble’. Such an idea captures the cultural aspects of retrospective explanations and predictions of disease and death, as well as assessments of risks, which sometimes also prove wrong [20]. Understandings of candidacy are reflected in how people enact and talk about the risks of disease and how they act on public health recommendations. Although Davison et al.’s [20] study context differs from that of the present study, we believe that their conceptualisation is highly relevant today because it acknowledges the complexity and sophistication of lay reasoning. Davison et al. [20] studied lay people’s understanding of the risks of coronary disease, while cervical cancer is the focus of the present study. The current social context in which health, risks, and disease are played out also differs from their study. Over the last three decades, the internet and social media have revolutionised access and exposure to (health) information, i.e., access and exposure to divergent and competing truths, narratives, and values.

Although often downplayed in public settings and media reports, there is a theoretical consensus that uncertainties characterise all aspects of medical and scientific practise. For instance, they are played out and managed in diagnostic practises in clinical settings [24]. In recent decades, the implementation of evidence-based practise has aimed to eliminate uncertainties within clinical settings by creating predictability, accountability, and streamlining processes using methodologies based on systematic reviews and meta-analyses of the best available research [25,26]. This shift clearly parallels the way in which questions of uncertainties in screening programmes should be managed [27], which, for instance, are related to whether screen-detected cancers progress to become clinically detectable or whether a person would die from another cause before the cancer becomes evident. Thus, an adverse consequence of screening is often termed ‘overdiagnosis’ or ‘overdetection’ [28], which for women involves the adverse consequence of overtreatment, which may be harmful [29].

Linking this framework to lay understandings of cervical screening may lead us to both uncertainties and ambiguities, which may reflect uncertainties in scientific epidemiological knowledge. However, of more significance is the idea that knowledge of these uncertainties and ambiguities may lead us to culturally accepted ways of dealing with health risks. Moreover, understanding the ways in which cervical cancer and its prevention are valued and ordered by lay people may provide new and important insights into the relevance and uptake of preventive health measures.

### 1.2. Aims

Our point of departure is an exploration of how cervical cancer is interpreted and understood during discussions involving women who do not comply with public health advice on cervical cancer screening, as well as the cultural beliefs, social environment, and political contexts these understandings seem to be entangled with. More specifically, the aim of this study is to investigate lay understandings of the epidemiology of cervical cancer and its prevention, specifically those discussed by women who are late for cervical cancer screening.

## 2. Materials and Methods

### 2.1. A Focus Group Study with a Social Constructivist Approach

We conducted a qualitative focus group study among women who were overdue for cervical screening in Norway to gain an understanding of nonattendance for cervical cancer screening in Norway. An analysis focusing on barriers related to booking a screening appointment was previously published [13]. In the current article, we focus on lay understandings of the risks and prevention of cervical cancer as reflected in focus group interviews (FGIs). We conducted nine FGIs with women who were overdue for cervical cancer screening. Focus group data might be understood as social enactments that involve integrating both interaction and content into the analysis [30,31]. This approach implies a social constructivist approach, within which FGIs are viewed as social arenas for the negotiation of social norms and normativity [32]. Applying this approach to our study, we understand FGIs as social arenas for negotiating socially acceptable ways of assessing and dealing with health risks and uncertainties.

### 2.2. Participants and Recruitment

The participants had not attended screenings in at least four years and were hence at least one year overdue. They were recruited through the NCCSP, which holds complete records of all cervical screening tests in Norway. Women living in Oslo (capital) or Finnmark counties who were reminded to attend screening in October and November 2017 or February 2018 received an invitation to take part in the study as an addition to the ordinary NCCSP reminder letter. We invited women living in the two regions because the screening participation rate was the lowest in these two counties. In addition, the counties represent extreme points in terms of geography and demography. Finnmark is sparsely populated and located in the northernmost part of Norway. The invitation included information stating that we would contact them by telephone to inquire about their possible participation in the study. It was made clear that participation was voluntary and that the invited women could easily opt out by declining our request. The first author and other research assistants phoned about 700 invited women. The majority could not be reached by phone. In the conversations with the women we reached, the researchers provided information about the study and outlined the issues we wished to discuss in the FGIs. A total of 65 women agreed to participate and were scheduled for the FGIs. Recruitment and interviews were carried out from November 2017 to May 2018. Five of the FGIs involved women who had not attended screenings for at least seven years. In total, 41 women attended and participated in the FGIs. Thus, 24 of the women who were scheduled for an interview failed to attend. After the FGIs, the participants were asked to fill out a short questionnaire on their background and knowledge of cervical cancer. Seven of the 41 participants had not heard of HPV before the interview. Ten were single/divorced, and 31 were married/in a relationship. Ten were outside the workforce (retired or on disability benefits). All participants were covered by universal health care insurance. The participants were between 29 and 69 years old. Three had immigrant backgrounds. Table 1 shows the socioeconomic/demographic composition of the sample.

To our knowledge, none of the participants had been vaccinated. The coverage of opportunistic HPV vaccination in this age group is very low [33]. A few stated they had daughters who had been vaccinated as part of the immunisation programme and therefore had some reflections about that.

### 2.3. Generation of Data

The FGIs were carried out in central Oslo in the facilities of the Norwegian Cancer Society or in a community health care centre in Finnmark. The FGIs were moderated by the first author. The last author assisted in five FGIs, and the fourth author assisted in one. The FGIs were conducted in Norwegian. The quotes presented in the present paper have been translated into English. Each FGI lasted about 90 min.

In the FGIs, the participants discussed several aspects related to cervical screening non/attendance. In line with Kitzinger’s [34] advice to maximise interactions and discussions in FGIs, we presented the participants with a few statements.

The statements triggered discussions on several topics related to screening attendance. The statements were based on previous research about barriers to screening attendance or were more directly related to the Norwegian screening context. In the last three focus groups, the issues were addressed with more open questions, such as ‘Why do you think women postpone the pap smear?’ In addition, we included three new statements to gain more specific insights into values, interpretations, and potential uncertainties associated with knowledge about cervical cancer. We prompted reflections on general attitudes and behaviours, as well as personal experiences and reflections related to cervical cancer epidemiology and screening. We continuously processed the interview content and concluded that the data had sufficient information power [35] after nine FGIs. The FGIs were digitally recorded and transcribed verbatim by the first author. To ensure anonymity, personally recognisable information was de-identified in the transcriptions.

### 2.4. Data Analysis

Based on our theoretical perspective, which directs attention towards uncertainties and ambiguities in scientific epidemiological knowledge as well as culturally accepted ways of dealing with health risks [32,34], our analytical focus was on *how* the epidemiology of cervical cancer and associated uncertainties were discussed in the FGIs. The first and last authors initially performed a joint reading of the interview transcripts, aiming to identify patterns across all the data. The analysis was continuously informed by our chosen theoretical perspective [36]. In particular, three themes running through the transcripts stood out as relevant for illuminating our research question: (1) aetiology and risk factors of cervical cancer; (2) cervical cancer screening; and (3) responsibility in acquiring cervical cancer. We worked in greater detail with a few sequences particularly related to these themes to unpack the complexity of the lay perspective. The interactive associations between different knowledge, interpretations, and uncertainties reflected in these sequences from the FGIs reflect distinct aspects of the lay epidemiology of cervical cancer. The first and last authors performed the analysis, while all authors contributed insights and questions to unpack the lay perspective reflected in the FGIs.

To provide an understanding of the actual interactions as they unfolded in the FGIs, we present sequences of discussions for each theme in the Results section. To make the discussions presented easy to follow, some of the original discussions were simplified (e.g., if several people talked at the same time or repeated themselves). Nevertheless, in the excerpts displayed in the Results section, we aimed to keep the meaning and tone as genuine as possible.

### 2.5. Approvals, Privacy, and Consent

The study received approval from the Privacy Ombudsman at Oslo University Hospital (ref. 2017/10642). All participants received an information letter outlining the aims and purpose of the study, the interview procedure, privacy, contact details, and the right to withdraw from the study at any time without consequences. All participants gave written consent to participate in the study before the interviews started. The data were stored on a server for sensitive data in compliance with the National Data Protection Guidelines. Participants were compensated €50 for expenses associated with attending the interview.

## 3. Results

### 3.1. Aetiology and Risk Factors of Cervical Cancer

Multiple understandings of certain risk factors were introduced in the discussions, and one main finding is that the aetiology of cervical cancer is understood as highly complex, which in turn had a significant impact on their reasoning for the candidacy of cervical cancer. This aspect is illustrated in the discussion below:


*FGI 9, Moderator: Is it random who acquires cervical cancer? What do you think—is that something you agree with or not?*



*1: Yes and no. I have been introduced to the word ‘BRCA’. Thus, if you are a BRCA carrier, then it is heredity, and that is also the case with cervical cancer. Otherwise, I do not see a connection to anything in my lifestyle, which should indicate that…*



*5: There are some factors that protect you. A virgin has a lower risk of acquiring cervical cancer but a greater chance of acquiring breast cancer. I know that because when we were taught oncology and cancer in nursing school, I remember well what they said about cancer: ‘The prostitute is protected against breast cancer because she breastfeeds, whereas the Madonna is protected against cervical cancer’. Hence, that is why; it is related to the explanation of the HPV virus. Due to that, I believe it is not random; there are protective factors. Then there are factors you are predisposed for too. Genetic or virus, or at least that is how I remember it!*



*Moderator 2: Interesting metaphors. Is it true that women who have not had sex have a very low chance of acquiring cervical cancer?*



*5: Yeah, and I believe there have also been repeated chlamydia infections. There are several risk factors. There are several issues with sexual risk behaviours that, of course, increase the chances of developing cancer.*



*Moderator 2: What do you think is the greatest risk of developing cervical cancer?*



*6: I believe it is heredity, but that is perhaps just something I believe.*



*3: Yeah, I believe that too.*



*6: I do not know if there’s been much research about that.*



*3: I also think of genetics and heredity, but it may be probable.*



*5: It is probably smoking and drinking!*



*Several laughing*



*3: Yes, maybe lifestyle. That may influence it.*



*5: Lifestyle, smoking, drinking, and wild, unrestrained, unprotected sex!*



*3: However, it is a bit random too, I believe; it is not just heredity. It may affect people a bit randomly.*


Interestingly, even though HPV is introduced and referred to, other risk factors and explanations seem far more dominant during the discussions of cervical cancer aetiology. The risk factors for cervical cancer are mainly associated with the risk factors for acquiring cancer in general. The starting point of this sequence, ‘I have, in a way, been introduced to the word “BRCA”’, reflects an understanding of the genetic predisposition for cervical cancer. Heredity is even reintroduced in the discussion after the elaborative account of the significance of HPV as a risk factor, thus adding weight to the understanding of aetiology as a complex of several risk factors.

Lifestyle as a risk factor is also introduced and included in the discussion: ‘Lifestyle, smoking and drinking, and wild and unrestrained, unprotected sex!’ The discussion indicates the difficulty of imagining health risks without including the significance of lifestyle in terms of optimisation of body, activity, and diet, as well as ‘responsible’ sexuality. Interestingly, drinking and smoking are introduced with humour, which indicates shared cultural ambivalence towards compliance with healthy lifestyle recommendations. This finding is in line with Davison et al. (1991), who argue that the idea of candidacy is often attended by laughter; that is, using humour may defuse danger and introduce the ‘unthinkable’ in everyday discourse. It also helps address the issue in a non-moralising way, i.e., implicitly addressing challenges with compliance but also reflecting a certain cultural resistance to the idea that maintenance of health can be reduced to a question of rational behaviours, choices, and control.

Another dimension that is introduced in the discussions of risks is the presumption that research evidence on cervical cancer aetiology is low, indicating that the aetiology of cervical cancer, to a great extent, is still ‘a mystery’: ‘I do not know if there exists research on this matter’. Thus, the notion that scientific knowledge may be incomplete and the occurrence of a general over-interpretation of the lack of scientific knowledge about cervical cancer aetiology also form part of the lay understandings reflected in our study. At the very end of this excerpt, the understanding of cervical cancer as randomly acquired is reintroduced: ‘but it i a bit random too’. Considering the previous discussion about HPV, heredity, and lifestyle, this statement might be interpreted as underscoring a perception of persistent uncertainties in the aetiology of cervical cancer. Remarkably, this strong notion of randomness corresponds to a scientific uncertainty about carcinogenesis and the probable element of luck as to why some people develop cancer while others do not. However, it simultaneously contrasts with the notion that risks and preventive factors have been extensively analysed and are well quantified.

Seeing this conversation as a whole, the focus gradually shifts from enacting associations with ‘protective factors’ to ‘risk factors’. As for the former, being a virgin is regarded as a protective factor, whereas sexual risk behaviour in terms of several sexual partners, unprotected sex, and repeated chlamydia infections is understood to increase the risk of acquiring cervical cancer. Thus, the risks of acquiring HPV and the risks of acquiring cervical cancer are not differentiated in the discussions.

### 3.2. Cervical Cancer Screening

In the previous section, when discussing the aetiology and risk factors of cervical cancer, the participants did not make explicit associations with screening attendance. Nonetheless, in discussions where screening and the purpose of screening were addressed more directly, several interpretations and associations with screening were articulated. The relevance of screening was often framed within a wider social context, clearly pointing to feelings of individual responsibility for attending screening; however, uncertainties associated with screening and diagnosis also surfaced.


*FGI 8, Moderator: Let us elaborate a little on the issue of taking the smear. Have you reflected on why you were asked or encouraged to take the smear?*



*5: It is preventive. I believe that is why. I believe it is safe to do it because then you may detect it early if it turns out you have it…*



*3: I believe that for the social and economic part, it would have been incredibly sad if so many, such a high share of women, would be hit by certain cancer types that can be prevented. Both treatment, being a patient, and losing the adult generation and everything…*



*4: But at the same time, I have also read a bit about the tendency to over-treat certain things. When you mass screen for different things, there are quite a lot of false-positive cases, and then you treat things that should not have been treated. It is a kind of waste of resources, perhaps giving people unnecessary anxiety. It is kind of a balance. It may apply more to breast cancer, perhaps because I have read about it or…*



*7: Small lumps that do not mean anything.*



*1: Yes, because it is a balance with regards to how many incidences per year they have before they start screening. Where do they draw the line for how much they should examine? Everyone can fall ill. For my own part, I believe it is [screening] prevention, just privately for my own part. Yeah, I believe that cancer may hit me. It is everywhere. I have many friends who think that ‘I cannot be bothered’ or ‘it will not happen to me’.*


In this sequence, the purpose of screening is reflected upon in several entangled ways, pointing in different directions. It is linked to both preventing and diagnosing cancer, particularly emphasising the diagnostic value of screening to start early treatment. In the following, screening is contextualised in terms of economics, which indicates a socio-economic legitimacy of screening: saving costs associated with treatment, human suffering, and early deaths is understood as a societal benefit of screening and a positive aspect of preventative medicine. To add weight to such logic, the idea of the burden of disease without a screening programme is emphasised. Such an understanding has implications for interpretations of screening attendance as not simply a personal choice relating to one’s own interests but rather as a social duty involving personal responsibility to stay healthy and promote personal health.

Looking even deeper into the uncertainties and ambiguities reflected in this sequence, cervical cancer screening is also associated with the controversy of overdiagnosis and the problem of false-positive cases. These issues, in turn, call into question the quality and efficiency of cervical cancer screening. The same arguments are used to articulate the imaginable negative aspects of screening as previously introduced unambiguously as the benefits, such as pointing out the societal costs related to medical overdiagnosis and the treatment of healthy women, as well as the suffering brought on by inflicting anxiety on women. Thus, the uncertainties with regard to both the extent of the overtreatment of healthy women and the extent of false positive tests in cervical cancer screening connect to a fundamental question within preventive public health about to what extent the purpose of prevention legitimises the harms it may also inflict.

Towards the end of the excerpt, another question is posed, namely, whether or not it is reasonable to start screening when there is a low incidence of the disease, which implies that the preventive effect would be insignificant if the disease is rare. This form of reasoning points to uncertainties with regard to how effective screening actually is. In the following, another question is posed: ‘Where do they draw the line for how much they should examine?’ The idea of such a ‘line’ reflects that the legitimacy of screening is low when the total number of prevented cases is presumed to be low. Thus, the acceptability and legitimisation of screening are up for discussion. On the one hand, the controversy isframed as critical reasoning about medicalisation, including reflections about both the drawbacks and health benefits of preventive medicine. On the other hand, it is addressed as a highly existential matter involving accepting the threat of disease as an inevitable part of life through statements as ‘it may affect me’. Such attitudes underscore a common understanding that it is impossible to control diseases. Thus, it stands in contrast to such attitudes as ‘it will not happen to me’. Nevertheless, both accepting the threat of disease and ignoring or distancing from the threat of such a disease may reflect conventional cultural attitudes regarding candidacy for cervical cancer, which may influence their perceptions of and dealings with health risks.

### 3.3. Responsibility for Acquiring Cervical Cancer

The third main topic of the discussions consistently revolved around the degree of individual responsibility for staying healthy and, hence, not acquiring cervical cancer. Such discussions implied that the candidacy of cervical cancer was framed by a wider social context, including entangled reflections on the social status of the disease, the moral aspects of acquiring HPV, and sexual relations as a risk factor. The following excerpt demonstrates these intertwinements:


*FGI 5, 7: It is a little taboo with cervical cancer because, you know, breast cancer has nothing to do with sexuality and that [HPV] transmits through sexual relations, and therefore it becomes a little more taboo.*



*Moderator: […] Do you think it is taboo in the sense that less is said about it, or that people do not know so much about it?*



*7: People may not want to be told that they have the virus.*



*1: I do not really know how credible it is, but I read an article about what status the different cancer types have. Breast cancer carries a higher status, while lung cancer, where you think that you are to blame for acquiring it, has a lower status. I can imagine that, yeah, if you think that it is sexually transferable, then it is like, ‘yeah, you should have known better.’ Maybe people think you are to blame and that you are responsible for acquiring cancer yourself. I do not know.*



*3: I think that is a very interesting issue. On the one hand, people are encouraged to have children, and having sex is such a natural thing that getting infected is very common. On the other hand, people feel responsible and shameful [for having HPV] since it is transmitted sexually: ‘No, I have inflicted it on myself’. I think it is not right if that is an attitude and a taboo for that reason.*



*2: I do not know why.*



*7: But I believe that it is as you say that it is high status and low status, but lung cancer and colorectal cancer are not just self-inflicted.*



*3: That it is caused by lifestyle.*



*2: Are not very, very many diseases related to lifestyle? And what do we do and do not do? In some way or another, diseases like diabetes, obesity, heart disease, and several others are associated with lifestyle. Thus, if we are to sit and not dare to talk about being ill and feel guilty for all the diseases we acquire, it gets complicated.*



*3: Yeah, they say, ‘live life while you can’, and that becomes really important here.*


The discussions above show that understandings and notions of cervical cancer, as with all diseases, are embedded in specific and changeable cultural frames. Especially the extent to which the disease is regarded as self-inflicted and controllable generates ambiguous meanings about acquiring it. As for the first aspect, self-infliction or not, the discussion demonstrates that cervical cancer is clearly interpreted in relation to other cancers, which are assumed to differ in status. Self-infliction of disease, for instance, related to smoking, is related to a low status of disease. In the discussions, cervical cancer is implicitly placed in the category of low status due to the sexual transmission of HPV. Thus, the negotiation of the status of cervical cancer seems related to whether or not individuals could be regarded as responsible for acquiring the disease. Thus, by referring to a moralistic perspective implicitly understood to be at work in society more generally, the disease may be interpreted as a sign of personal failure to take responsibility for staying healthy through the exercise of responsible sexual behaviour. Interestingly, though, we found that the moral judgement related to self-infliction was not associated with screening non-attendance. This finding raises a question about whether lay understandings of the significance of lifestyle and health behaviours overshadow other forms of preventive health behaviour, such as screening attendance.

Another dominant and critical point of view running through this discussion was on the consequences of carrying a moral responsibility for staying healthy and a particular concern about the stigmatisation of women acquiring cervical cancer. This concern might be the background for the reasoning towards the end of this sequence, wherein alternatives to understanding risks as beyond moral and personal responsibility are also imagined. The reflections indicate that fatalistic values are understood as necessary in order to cope with an overwhelming focus on moral responsibility for one’s personal health. Accordingly, a ‘rational’ way of dealing with the illness is to ‘live life while you can’. The context for this statement was not a response to how to deal with the threat of disease, but how to deal with the potential moralisation and stigmatisation of disease. Such a context might make sense of the articulations that devalue efforts to control the disease and of public health messages that may be interpreted as moralistic.

## 4. Discussion

The present study shows how lay understandings of the epidemiology of cervical cancer and its prevention were discussed among women who were late for cervical cancer screening. Several aspects of epidemiology are interpreted and used in negotiating the candidacy of cervical cancer, i.e., acceptable and nuanced understandings of the epidemiology of cervical cancer. In particular, understandings of acquiring cervical cancer as beyond individual (and to some extent medical) control and of personal responsibility for acquiring cervical cancer were brought up for discussion.

This study’s findings provide insights into how understandings of the candidacy of cervical cancer articulated by women late for screening may reflect a larger social context in which screening is given less relevance or significance than aimed for. The findings resonate with other recent studies showing that despite the clear public health strategy to eliminate cervical cancer worldwide through organised screening and vaccination programmes [37], screening as prevention is not necessarily perceived as relevant at an individual level [11]. In fact, in the UK as many as 80% do not associate screening with prevention [38]. Hence, screening attendance has been understood in terms of cultural norms, such as normative expectations of sensible and responsible behaviours or a moral obligation and female duty signifying good and responsible citizenship [39,40,41].

Previous surveys have indicated low levels of awareness and knowledge about HPV among the general public [42,43]. In our study, it became clear that although HPV was vividly discussed and interpreted and the association between sexual behaviour and cancer risk was explored rather thoroughly, the candidacy for cervical cancer was understood as a more complex matter than simply HPV infection causing cancer. In the FGIs, additional etiological explanation models were given equal or even greater relevance than HPV to explain the causes of cervical cancer. Hereditary factors were interpreted as very important for the causation of cervical cancer—a notion not extensively investigated or supported in previous studies [44]. Thus, genetics serves as a very powerful and relevant discourse that adds to the uncertainties regarding the causation of cervical cancer, particularly when it comes to the limitations of individual control over disease causation. This finding resonates with former studies, which argue that heredity is high up in the explanatory hierarchies of the causes of cancer [22]. Other powerful explanatory models of risk factors introduced in our study were clearly related to health behaviours, which included both lifestyle in broad terms and sexual risk behaviours. It has been well-documented in the literature that people tend to associate lifestyle with the causation of disease [18,21,45], which was also the case in this study. Thus, the weight of heredity and lifestyle in understandings of risk factors may provide insights into contexts in which screening is given a more relative and uncertain value.

In our study the candidacy of cervical cancer was also reflected through fatalistic statements on bad luck and randomness, which, according to Davison et al., are highly characteristic of lay perspectives on epidemiology [21]. It highlights candidacy as fallible and may help account for unwarranted survivals and anomalous deaths [20]. Interestingly, fatalism is often viewed as a cultural belief and consequently contrasted with scientific evidence [46,47]. However, in the current context, a notion of luck also corresponds to general scientific uncertainty about carcinogenesis, along with the element of randomness being linked to acquiring cancer caused by random mutations arising during DNA replication in normal, non-cancerous stem cells [48]. In the context of HPV, there is still scientific uncertainty regarding its immunology and why it goes on to cause cervical cancer in only a fraction of the women who are infected by this very common virus [49].

During the FGIs, the participants typically did not draw simple conclusions when discussing the epidemiology and prevention of cervical cancer but rather displayed complex and entangled associations with the candidacy. In fact, the uncertainties articulated helped expand understandings about what can be considered rational health behaviours. Furthermore, such uncertainties added weight to a sense of lack of control and predictability over disease and made the personal and moral responsibility of staying healthy problematic. In our study, the candidacy of cervical cancer was also reflected as an idea they were concerned about and opposed due to the role of HPV as a sexually transmitted virus. Thus, a well-known discourse dichotomising women’s sexuality into good or bad [50] was contested, with a specific concern about the moralisation of sexual behaviour, the stigmatisation of cervical cancer, and the moral judgements of women acquiring the disease also highlighted in other studies [16,50]. Ambivalence of personal responsibility and opposition to the moral or stigma attached to the disease may reflect a broader ambivalence towards a ‘Western’ public health ideology that endorses personal responsibility for health, which is typically achieved through discipline, self-control, and adherence to a healthy lifestyle [51]. Thus, such ambivalence reflects the unintended consequences public health recommendations may be interpreted to have, i.e., to underpin stigmatisation and moral judgement about acquiring certain diseases. This, in turn, opens up alternative ways of taking care of one’s health, as represented by the saying, ‘live life while you can’. This may constitute another context in which screening may be rendered less relevant or significant for maintaining health.

### 4.1. Implications

Implicit in much public health communication is the assumption that the more knowledge people have, the more ‘rational’ they become and, consequently, the more compliant they are with health recommendations. According to Allmark and Tod [52], public health is challenged when the public doubts or does not act on public health messages. In our study, lay understandings among women who are late for screening came out as complex understandings with inherent uncertainties. Such complexities were set in opposition to moralistic public health messages interpreted by the participants to be at work and made up of several sources of information. This result adds insights into why rationalised and simplified public health messages may not be interpreted as plausible and, consequently, may result in scepticism or be rendered counter-productive [21,53]. Public health messages regarding the prevention of cervical cancer may obscure the uncertainties inherent in its related practises and ignore the specific cultural judgements and values that are ultimately inextricable from such messages [53]. Moreover, the perceived emphasis on lifestyle through public health messages may, perhaps, overshadow other public health messages, such as screening as a preventive measure. In line with Allmark and Tod [52], we argue that by unpacking the cultural contexts within which people make sense of health and disease, lay epidemiology can enhance the relevance and effectiveness of public health guidance and shape future cancer prevention programmes.

### 4.2. Methodological Considerations

This study is innovative because the recruitment procedures we used involved the registries to identify the target group (i.e., women who had not complied with current recommendations for screening). The invitations were sent out in collaboration with the NCCSP, which made it possible to specifically target women who were overdue for cervical cancer screening.

In the FGIs, the researchers aimed to facilitate open discussions, which allowed the participants to communicate their immediate reflections. The first and last authors, both female sociologists, developed the interview guide and carried out the FGIs. Indeed, this background made them particularly attentive to how understandings of cervical cancer epidemiology reflect cultural interpretations and structural conditions.

Despite the high number of participants in this study, most of the women we contacted did not participate, either because we were unable to reach them or because they simply did not want to take part in the study. In addition, 24 of the women who had expressed a wish to take part in the study failed to attend the scheduled FGIs. This made the number of participants in each FGI highly unpredictable. In two of the FGIs, only two participants showed up. This limited the discussions and interactions to some extent since fewer viewpoints, attitudes, and reflections were shared. In addition, the sample of women who took part in this study reflects self-selection bias. Most participants had relatively high levels of education (3-year college degree or 5-year university degree). Some of the participants were also health care professionals. This may partly explain the relatively high degree of health literacy, which was reflected in some FGIs and added to the complexity of the lay epidemiology of cervical cancer. Therefore, the sample does not reflect the general population of non-attenders, in which women with immigrant backgrounds and women outside of the workforce are overrepresented [54]. A more heterogeneous sample may have added other dimensions to the discussion. However, the sample’s relatively homogeneous composition may have facilitated exchanges of (anticipated) widespreadcultural views, including both fatalism and critical interpretations of scientific evidence.

The focus group methodology generated an understanding of how the lay epidemiology of cervical cancer consists of multiple and divergent views of epidemiology and its prevention. Individual interviews may, to a greater degree, yield insights into how lay epidemiology is constructed into more coherent individual understandings [23].

## 5. Conclusions

Medicine has been criticised for not recognising the sophistication of lay understandings of risks and the prevention of disease. Hence, an innovative methodological and theoretical approach was developed to generate an understanding of lay reasoning about the explanations and predictions of cervical cancer. By investigating how women who were late for screening discussed the epidemiology of cervical cancer and its prevention, this study provides novel insights into the ways in which different associations are used in negotiating acceptable and culturally adaptable understandings of the candidacy for cervical cancer. An important aspect of such negotiations was the reinforcement and enactment of uncertainties about risk factors for acquiring cervical cancer. Such uncertainties served an important function of bringing up for discussion specific understandings of how cervical cancer is acquired as being beyond one’s individual control and personal responsibility. Hence, lay epidemiology should not be dismissed but rather considered to provide an enhanced understanding of the contexts in which the relevance of screening and decision-making with regard to screening attendance are played out.

## Figures and Tables

**Table 1 healthcare-11-01441-t001:** Characteristics of the participants in the focus group interviews.

		Education Level			Age				Immigration Status	
	Attend. Status	Secondary	Lower Degree (College)	Higher Degree (University)	29–39	40–49	50–59	60–69	Born in Norway	Born outside Norway
FGI 1 (n = 6)	>4 years	2	2	2	1	1	3	1	6	
FGI 2 (n = 5)	>4 years	2	2	1	3	1		1	5	
FGI 3 (n = 6)	>4 years		1	5	4		2		6	
FGI 4 (n = 2)	>7 years	1		1			1	1	2	
FGI 5 (n = 4)	>7 years	1	2	1			2	2	2	2
FGI 6 (n = 2)	>7 years		1	1	1	1			2	
FGI 7 (n = 3)	>7 years		2	1		1		2	2	1
FGI 8 (n = 7)	>7 years	1	3	3	1		3	3	7	
FGI 9 (n = 6)	>4 years	1	2	3	2	4			6	
Total (N = 41)		20%	36%	44%	29%	20%	27%	24%	93%	7%

## Data Availability

No data are available.

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
