# Peer review of "Unpacking the Lay Epidemiology of Cervical Cancer: A Focus Group Study on the Perceptions of Cervical Cancer and Its Prevention among Women Late for Screening in Norway"

_healthcare, 2023, doi:10.3390/healthcare11101441_

Round 1

Reviewer 1 Report

The study was well designed and written up. I would suggest adding the two regions participants were recruited from (Finnmark and Oslo) to the breakdown in Table 1.

Author Response

Dear referee

Thank you for your insightful comments and for your efforts in improving our paper. Below we give an account of how we have dealt with each comment. Substantial changes has been marked up in yellow in order to make it easier to track and evaluate our revisions. We hope we have adequately addressed your concerns. The manuscript has been proof read by a professional proof reader.

Referee 1

The study was well designed and written up. I would suggest adding the two regions participants were recruited from (Finnmark and Oslo) to the breakdown in Table 1.

Response: We have discussed this suggestion. Due to concerns of the participants’ privacy/anonymity, we have concluded that it is not advisable to include information in the table about what region the interviews were carried out in.

Reviewer 2 Report

The paper reports the results of several focus groups of non participants in cervical cancer screening in urban populations in Norway. The perspective of the authors is very interesting and potentially useful to understand the lack of participation in screening. The methodology is well suited for the objective.

I have a couple of comments on the paper:

- it seems clear that the participants were the same than in a previous paper published on BMJ Open. Although the analysis and results published are different, I think it should be explicitly acknowledged in the paper. Also, the contribution of the previous paper may be included in the introduction and explain why this paper is a new analysis.

- it would be useful to know how many women refused to participate and how many were not reached.  Also, how many accepted but they did fail to attend (avoiding expressions such as 'about 1/3')

Author Response

Dear referee

Thank you for your insightful comments and for your efforts in improving our paper. Below we give an account of how we have dealt with each comment. Substantial changes has been marked up in yellow in order to make it easier to track and evaluate our revisions. We hope we have adequately addressed your concerns. The manuscript has been proof read by a professional proof reader.

Referee 2:

Referee: …….It seems clear that the participants were the same than in a previous paper published on BMJ Open. Although the analysis and results published are different, I think it should be explicitly acknowledged in the paper. Also, the contribution of the previous paper may be included in the introduction and explain why this paper is a new analysis.

Response: We have included a more explicit acknowledgement of the paper and an elaboration of how the two analysis/papers differ in the “materials and methods section” (see p. 6, line 7-10). The paper published in BMJ Open is also cited in the introduction.

Referee: It would be useful to know how many women refused to participate and how many were not reached. Also, how many accepted but they did fail to attend (avoiding expressions such as ‘about 1/3’)

Response: We acknowledge the usefulness of providing information about how many of the participants refused to participate and how many were not reached. Unfortunately, this information was not recorded during the recruitment process, although the impression was that the majority was not reached. We have written more explicit how many accepted but failed to attend in the ‘participant and recruitment section’, p. 7, line 12-13.

Reviewer 3 Report

Healthcare-2222043

Unpacking the lay epidemiology of cervical cancer: A focus 2 group study on the perceptions of cervical cancer and its prevention among women late for screening in Norway

This is a well written and interesting paper. Minor comments.

Abstract

Results: the results in the paper do not mention several biomedical explanation models.  There is also limited discussion around risk. Reconsider what is summarised in the results section of the abstract to reflect what is described in the paper. Also perhaps a brief introduction of the term candidacy

Conclusion: its not linked to the results. As the results do not discuss “cultural perceptions and values”. The paper does in detail.

Background

For an international audience it would be important to know the screening interval and if this changes if HPV is detected.  If HPV is detected what happens next? I am assuming cervical screening includes HPV testing if not please state this.  Also is cervical screening done by health care professionals or can women self-collect their sample which is now common in several countries to remove the access barriers?  This background information is important to understand the context.

Methods

Minor comment the data were collected in 2017-2018 (pre Covid) almost 5 years ago. Can it be assumed that the screening rates are even lower due to Covid? Have there been any changes to the screening program since then?

Participant characteristics: perhaps not part of the data collection but if there information the percentage of women who had an HPV vaccine? Especially considering this will impact on their perceptions of risk and their views of screening.

Data collection: the FG discussions were conducted in Norwegian and we can infer data analysis was also conducted in Norwegian and only the quotes were translated.

Focus groups: considering the nature of the topic did the authors consider individual interviews allowing women to provide more in-depth description of the reasons why “they” didn’t attend screening. I can understand the richness of the interactions but it also shows that women did not talk about themselves in first person i.e I decided not to screen because. Having said that I understand the aim with the lay meaning not the barriers to screening.

Results

While its hard as these were FG discussions where there differences based on participants age? As above perceptions could have been influence by HPV vaccination available to the younger cohort of women.

As mentioned in the limitation section the participants are highly educated but some also appear to have health literacy as the terms “incidence” and  “prevalence” (not a common lay term). Is used.

Discussion

Page 10, 1st paragraph the authors state: “the weight of heredity and lifestyle as risk factors may partly explain why screening non-attendance as a risk factor within lay epidemiology is given a more relative and uncertain value”.  Not according to the results. It is still unclear why these women did not participate in screening as the focus was on their understanding not on the barriers to screening.  

Author Response

Dear referee

Thank you for your insightful comments and for your efforts in improving our paper. Below we give an account of how we have dealt with each comment. Substantial changes has been marked up in yellow in order to make it easier to track and evaluate our revisions. We hope we have adequately addressed your concerns. The manuscript has been proof read by a professional proof reader.

Referee 3:

Abstract, Results section: the results in the paper do not mention several biomedical explanation models. There is also limited discussion around risk. Reconsider what is summarized in the results section of the abstract to reflect what is described in the paper. Also perhaps a brief introduction of the term candidacy.

Response: We agree that the formulation do not reflect the results in the article well enough. Therefore we have changed the sentence and broaden the focus: “Both interpretations of biomedical concepts and cultural values were used to negotiate acceptable and nuanced interpretations of candidacy for cervical cancer”. In addition, we have included a brief introduction of the term candidacy in the abstract, as suggested (p.1, line 32-35). We also expanded some on the definition of candidacy in the introduction (page 4, line 13-14, 20).

Abstract (Conclusion): its not linked to the results. As the results do not discuss “cultural perceptions and values”. The paper does in detail.

Response: We acknowledge that the last sentence of the conclusion is a too general statement and is not well enough linked to the results. We have decided to drop this sentence.

Background - For an international audience it would be important to know the screening interval and if this changes if HPV is detected.  If HPV is detected what happens next? I am assuming cervical screening includes HPV testing if not please state this.  Also is cervical screening done by health care professionals or can women self-collect their sample which is now common in several countries to remove the access barriers? This background information is important to understand the context.

Response: Information about screening interval, sample collection and HPV testing is now included in the background section (see p. 2, line 19-25, p. 3, line 1-4, 7-9). In addition, the flow chart/algorithm of the Norwegian cervical cancer screening programme is included as a supplementary file. This provides an overview of how positive HPV tests are followed up.

Methods: the data were collected in 2017-2018 (pre Covid) almost 5 years ago. Can it be assumed that the screening rates are even lower due to Covid? Have there been any changes to the screening program since then?

Response: Updated information about the screening attendance during and after Covid is now included (p. 3, line 5-6). The implementation of HPV testing as primary screening has been ongoing the last five years and as we detail in the paper, since summer 2022 it has been implemented nationwide (p. 2, line, 19-22).

Participant characteristics: perhaps not part of the data collection but if the information the percentage of women who had an HPV vaccine? Especially considering this will impact on their perceptions of risk and their views of screening.

Response: To our knowledge, none of the participants had been vaccinated. The coverage of opportunistic HPV vaccination in this age group is very low (Dong et. al. 2021). A few told they had daughters who had been vaccinated as part of the immunization programme and had therefore some reflections about that (p. 8, line 2-5). After each FGI we asked all participants to fill out a simple questionnaire. The questionnaire also included information about whether or not they had heard about HPV before the interview. Seven of the participants had not heard about HPV before they participated in the FGI. We have included this information in the methods section where we describe the participants (p. 7, line 14-15). It is not possible to say how or to what extent this might have impacted their reasoning about risks and prevention of cervical cancer.

Data collection: the FG discussions were conducted in Norwegian and we can infer data analysis was also conducted in Norwegian and only the quotes were translated.

Response: This information has been included in the revised manuscript (p. 8, line 9-11).

Focus groups: considering the nature of the topic did the authors consider individual interviews allowing women to provide more in-depth description of the reasons why “they” didn’t attend screening. I can understand the richness of the interactions but it also shows that women did not talk about themselves in first person i.e. I decided not to screen because. Having said that I understand the aim with the lay meaning not the barriers to screening.

Response: We acknowledge this aspect of our choice of data gathering and have therefore included some further reflections on this matter in the “methodological considerations” section in the discussion, p. 22, line 10-14.

Results

While its hard as these were FG discussions where there differences based on participants age? As above perceptions could have been influence by HPV vaccination available to the younger cohort of women.

Response: To our knowledge, none of the participants had been vaccinated. The coverage of opportunistic HPV vaccination in this age group is very low (Dong et. al. 2021). A few told they had daughters who had been vaccinated as part of the immunization programme (p. 8, line 2-5). Seven of the participants had not heard about HPV before they participated in FGI. These women were spread across different age groups (p. 7, line 14-15).

As mentioned in the limitation section the participants are highly educated but some also appear to have health literacy as the terms “incidence” and “prevalence” (not a common lay term). Is used.

Response: 80% of the women had higher education (university or college degree), and some had health professional background which may also explain their familiarity with terms such as incidence or false positive cases. We have included further information about their professional background in ‘methodological consideration’ in the discussion (p. 22, line 1-4).

Discussion

Page 10, 1st paragraph the authors state: “the weight of heredity and lifestyle as risk factors may partly explain why screening non-attendance as a risk factor within lay epidemiology is given a more relative and uncertain value”.  Not according to the results. It is still unclear why these women did not participate in screening as the focus was on their understanding not on the barriers to screening.  

Response: We agree on this and have therefore clarified the sentence and changed it to: “Thus, the weight of heredity and lifestyle in their understandings of risk factors may provide insights into contexts in which screening is given a more relative and uncertain value”. Thus, the complex understandings of the lay people may provide insights into processes where public health recommendations are given relative or uncertain value, please see p. 19, line 2-4.
